# Incidental Data: A Survey towards Awareness on Privacy-Compromising Data Incidentally Shared on Social Media

Stefan Kutschera [1,*], Wolfgang Slany [1], Patrick Ratschiller [1], Sarina Gursch [1], Patrick Deininger [1,2] and Håvard Dagenborg [3]

1 Institute of Software Technology, Graz University of Technology, 8010 Graz, Austria; wolfgang.slany@tugraz.at (W.S.); patrick.ratschiller@ist.tugraz.at (P.R.); patrick.deininger@student.tugraz.at (P.D.)
2 Institute of Software Design and Security, FH JOANNEUM Gesellschaft mbH, 8605 Kapfenberg, Austria
3 Department of Computer Science, UiT The Arctic University of Norway, 9037 Tromsø, Norway; havard.dagenborg@uit.no
* Correspondence: stefan.kutschera@ist.tugraz.at; Tel.: +43-3168735745

**Abstract:** Sharing information with the public is becoming easier than ever before through the usage of the numerous social media platforms readily available today. Once posted online and released to the public, information is almost impossible to withdraw or delete. More alarmingly, postings may carry sensitive information far beyond what was intended to be released, so-called incidental data, which raises various additional security and privacy concerns. To improve our understanding of the awareness of incidental data, we conducted a survey where we asked 192 students for their opinions on publishing selected postings on social media. We found that up to 21.88% of all participants would publish a posting that contained incidental data that two-thirds of them found privacy-compromising. Our results show that continued efforts are needed to increase our awareness of incidental data posted on social media.

**Keywords:** incidental data; privacy; social media; awareness

## 1. Introduction

The reckless use of social media and other online services can expose insights into personal affairs beyond what was intentionally meant to be shared. The posted data might reveal significantly more information than first thought, particularly when analyzed and combined with auxiliary databases. For instance, it has been shown that algorithms can estimate a person's personality effectively from likes [1–3]. With regard to the rich multimedia data that are posted in large volumes on social media, one might suspect significant unintended disclosure of personal information amongst the many inconspicuous pictures and videos of family and friends. Such unintended disclosure of personal information can threaten someones privacy. For this reason, this study is of interest to investigate the awareness of a person posting content on social media. In this paper, we study the awareness of unintentionally published data on social media, commonly referred to as incidental data [4]. We conduct a survey that implicitly assesses the participant's awareness of incidental data without the influence of the question itself through a survey, as the mere assessment of a privacy concern increases its awareness [5]. As a methodology, we used a survey that asked questions regarding postings that contained incidental data. In our previous research [6], we analyzed postings using *Open Source Intelligence* (OSINT) methods while limited to two hours per target. It was possible to detect data that were not intended to be published within that time limit. The responses from the quantitative survey method are then analyzed using statistical methods. Further, data found within those postings lead

to further data that most found privacy-compromising. Our survey provides robust evidence that there exists a significant lack of awareness around incidental data in social media postings as more than one-fifth of participants would share content containing hidden data they find privacy-compromising. We have indicators that suggest awareness of hidden data that can be a threat to privacy; thus, the focus on proper guidelines and education to help prevent the self-publication of incidental data may be a point of improvement. This paper makes the following contributions:

- Presents a novel survey methodology that indirectly evaluates participants' awareness of incidental data, avoiding the influence of the question itself;
- Provides robust empirical evidence of the lack of awareness among social media users regarding the privacy implications of their online content;
- Highlights the prevalence of incidental data in social media posts and its potential threat to user privacy.

In the first part of this paper, we discuss the current state of the art and summarize the views on the psychological impact of disclosure or nondisclosure of information. Then, we provide insights on our study design and further list the questions of our survey, including answer choices. Next, we show and describe our results in Section 4 before we evaluate and discuss those results in Section 5. We summarize our study with a final conclusion in Section 6.

## 2. Background

Privacy on social media is often discussed in terms of either privacy-jeopardizing settings or malicious actors [7]. However, Krämer and Schäwel [8] discuss the urge of people to self-disclose personal information on social media. Schneier [4] defined a taxonomy of different data that is in connection with the usage of social media, namely, the following: service, disclosed, entrusted, incidental, behavioral, and derived data. In particular, Schneier [4] argues that incidental data in this context are data posted by other people, over which one has no control.

**Definition 1.** *"**Incidental data** is what other people post about you: a paragraph about you that someone else writes, a picture of you that someone else takes and posts. Again, it's basically the same stuff as disclosed data, but the difference is that you don't have control over it, and you didn't create it in the first place." Schneier [4].*

In previous work [6], we argue that the term incidental is used in case something unexpected was found that should not be there. For instance, during an X-ray examination meant to assess a potential bone fracture, the discovery of tumorous tissue is termed an incidental finding [9]. Considering this more general meaning, we argue that one's unawareness of unintentionally publishing problematic data, alongside the primary reason for publishing content on social media, also leads to the uncontrollability of personal data. Fitting the intent behind the definition by Schneier [4], we propose an extended Definition 2.

**Definition 2.** ***Incidental data** is data that one has no control over, either due to another person disclosing it or the unawareness of its existence within data disclosed by oneself.*

### 2.1. Privacy from a Psychological Perspective

As argued by Schlosser [10], *self-disclosure* can be defined as communicating personal information about oneself to another person that is a close representation of oneself. Whereas *self-presentation* is defined as controlled and directed information that impacts the impressions of people about oneself Schlosser [10]. Barasch [11] discusses *intrapersonal* as emotions and processes within oneself, whereas *interpersonal* describes effects on relationships between others and oneself.

Luo and Hancock [12] state that disclosure fulfills basic social needs and thus improves one's well-being. Krämer and Schäwel [8] continue that privacy is an intrapersonal secondary need for people. Equally important is the view of the intrapersonal and interpersonal cost of not disclosing information. Sharing a problem (privileged information)

can help to improve one's situation by gaining new views on a personal topic or the view of others about oneself [13]. However, the consequences of sharing information are often overestimated [13]. Furthermore, the sharing of secrets can be used in a strategic manner to evade criticism and gain support [14]. Consequently, it can be said that disclosing information can have positive effects and be vital for oneself.

Social media seems to have filled a perfect spot that can fulfill the human motivation for self-disclosure [8]. However, this also entails dangers, as interactions with social media as simple and trivial as giving *likes* to certain posts can give away personal information [1,15].

Brough and Martin [5] claim that research on privacy is strongly focused on a user's motivation to protect their personal data from unauthorized usage, which correlates to privacy concerns; however, they focus very little of their research on privacy knowledge. The authors further state that privacy concerns might be artificially increased when they are being assessed.

Automated data collection and the usage of specialized algorithms can reveal sensitive information about one's life [16,17]. Fast and Jago [16] find that people underestimate the risks of sharing personal data; moreover, people seem unable to take strong actions even after severe privacy violations [18]. Such behavior comes from focusing on benefits and convenience combined with not being an explicitly identifiable victim [19]. Conversely, the benefit and convenience of data collection and usage of algorithms pose a massive threat to one's privacy; however, this may have created a state of mind where people think it is not realistically possible to stop it.

## 2.2. Privacy from a Technical Perspective

When people interact with a social network of their choice, it can be assumed that the main goal is to share content and not to tackle a host of privacy settings. This can be problematic as companies have discovered that user data, especially of a large group of people, can be a valuable asset. Even though there are good examples of user-based privacy, there are companies that take advantage of people's behavior. As discussed by Bösch et al. [20], such methods are referred to as dark privacy strategies or dark privacy patterns. Research in human–computer interaction and user-experience design has found that people are more likely to press a button in a rush if it is green. This led to situations where companies made the accept button for "allow cookies" or "share statistical data" buttons slightly larger and green, whereas the decline button is slightly smaller and gray [21,22].

Al-Charchafchi et al. [23] found in their review that users are threatened in multiple ways. The threat vectors concern information privacy, social engineering, data leakages through unfit privacy settings, or *Application Programming Interface* (API) weaknesses. A similar line is taken by the work of Johansen et al. [24], with the authors providing an insight into the problems and opportunities of lifelogging systems. In forensics and in court, the analysis of *Electric Network Frequency* (ENF) becomes used more often in order to verify timestamps or the untampered integrity of audio and video recordings [25–27].

## 2.3. Privacy from an Awareness Perspective

The quantitative study of Amon et al. [28] on interdependent privacy provides valuable insights into aspects of privacy awareness, especially the sharing of private information of other persons. The study analyzed 245 responses on 68 real-world pictures out of 13 categories through a questionnaire about the likelihood of sharing given pictures, entertainment, and its privacy rating. The study assessed the specific personality traits known as the *dark triad*, which focuses on narcissism, psychopathy, and manipulative personality style. Even though the study gives valuable insights into privacy awareness on social media, it focuses on pictures shared by others. Based on the responses on the pictures and personality traits, the findings from a cluster analysis were the following three interdependent privacy user categories: privacy preservers, privacy ignorers, and privacy violators. The study reveals that privacy ignorers have a low dark triad and low levels of education level but prefer personal privacy. Privacy violators have a high dark triad, high levels of education, and

further prefer openness as a key motivation factor for sharing potentially sensitive pictures of other persons.

Padyab et al. [29] conducted two sub-studies regarding privacy awareness on social media based on exploratory focus groups. The first tackles dedicated algorithms on social media; the second explores self-disclosure. These studies show that users were generally unaware of the extent published data can be used to extract private information. Further, it was shown that a user's awareness could be raised by letting them use an extraction tool on their own social media profile.

## 3. Implementation

We conducted the following four separate surveys: IDS2301, IDS2301U, IDS2302, and IDS2302U. It is important to mention that the presented study relies only on one survey, namely, IDS2301. Nonetheless, surveys IDS2301U, IDS2302, and IDS2302U were implemented in order to obtain reliable results, as explained in Section 5.4. An overview of each survey, including the motivation, can be found in Table A1. In accordance with the *General Data Protection Regulation (EU) 2016/679* (GDPR), an online survey provider was chosen in order to conduct the survey and collect responses. For the social media postings, we decided to use two well-analyzed postings from our previous work [6], while limited to two hours per target, Kutschera [6] found that privacy-compromising data unintentionally posted could be found by using OSINT. For the IDS2301 survey, we ran a test phase where 12 participants were asked to give feedback on the consistency, subjective understandability, and potential typos. The feedback was used to improve the survey. At the end of the survey, a link to the dedicated follow-up survey was presented.

### 3.1. Recruitment

Participants were recruited from three different courses at Graz University of Technology, Austria: *INH.04062UF Agile Software Development* (170 students), *INP.32600UF Mobile Applications* (45 students), and *INP.33172UF Software Technology* (84 students). Of all 299 students, 198 optionally participated in survey *IDS2301*, as shown in Figure 1. Students who were signed up for two or more lectures were only allowed to take the *IDS2301* survey once.

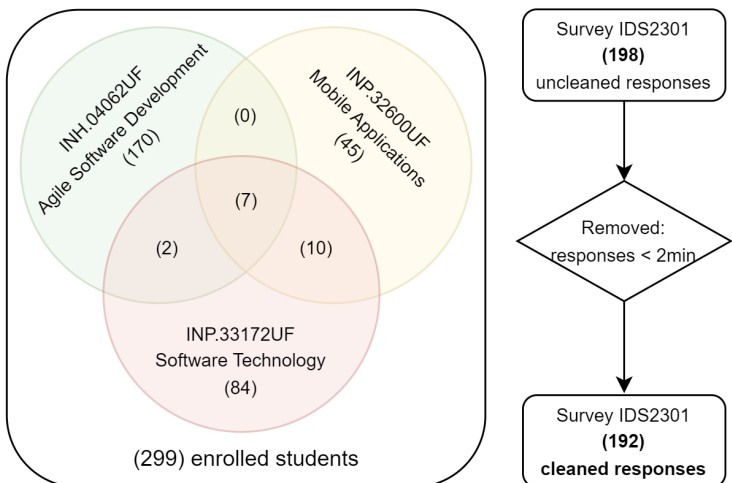

**Figure 1.** Venn-diagram of the number of enrolled students in the three courses we recruited participants from and how they overlap (**left**); and the number of responses before and after cleaning (**right**).

To motivate participation, students were offered bonus points counting toward their final grade for completing the survey. To claim points, a student had to submit a self-generated random *Universal Unique Identifier* (UUID) token as part of the follow-up survey and subsequently to the university's e-learning platform. Submitting the token as part of a separate follow-up study instead of the main study, allowed us to correctly identify students for the purpose of crediting points while at the same time preserving their anonymity in

main survey. Section 5.4 discusses this aspect in more detail. The instructions on how to claim points were only revealed at the end of the main survey, which reduced the risk of students going directly to the follow-up study to claim credits, skipping the main survey. It was certainly possible to receive these instructions out-of-band, circumvent the protection mechanism, and claim points without accessing the main survey. However, this was by design, as we found it preferable over the case where students would fast-click through the survey and submit bogus data. Our scheme gives no incentive to complete the survey multiple times, as bonus points will only be received once.

### 3.2. Assessment Design

In our survey, we judiciously employed 5-point and 6-point Likert scales [30] for distinct sets of questions driven by the nature of the responses we sought to capture. For 23 of our questions, we utilized the 5-point Likert scale, acknowledging its capacity to provide a balanced range of options from "Strongly Disagree" to "Strongly Agree," along with a neutral midpoint. Including a neutral option in these cases allows for a more accurate representation of respondent attitudes, particularly when they may lack a definitive opinion or possess moderate views [31]. This configuration was especially suitable for questions Q1 and Q2, as depicted in Table 1, and Q3.1 to Q3.21, as seen in Table 2, where neutrality or a middle-ground perspective was a plausible and informative response.

**Table 1.** Survey IDS2301 Questions 1–2.

| Reference | Question | Response Type |
|-----------|----------|---------------|
| Q1 | If this was a video from your house, would you share it publicly? | 5-P Likert Scale |
| Q1.1 | Explain your decision (optional) | open-ended |
| Q2 | If these were pictures from your house or surroundings, would you share them publicly? | 5-P Likert Scale |
| Q2.1 | Explain your decision (optional) | open-ended |

**Table 2.** Survey IDS2301 Question 3.

| Reference | Question | Response Type |
|-----------|----------|---------------|
| Q3 | Do you agree or disagree that the following data can compromise your privacy if shared incidentally on social media? | |
| Q3.1 | State or country you currently live | 5-P Likert Scale |
| Q3.2 | Your full name | 5-P Likert Scale |
| Q3.3 | Your date of birth | 5-P Likert Scale |
| Q3.4 | Your full address | 5-P Likert Scale |
| Q3.5 | Full name(s) of previous owner(s) | 5-P Likert Scale |
| Q3.6 | Your blood type | 5-P Likert Scale |
| Q3.7 | Your social security number | 5-P Likert Scale |
| Q3.8 | Information on relatives (name, date of birth or phone number) | 5-P Likert Scale |
| Q3.9 | Your phone number | 5-P Likert Scale |
| Q3.10 | Your email address | 5-P Likert Scale |
| Q3.11 | Your postal tracking number | 5-P Likert Scale |
| Q3.12 | Your parcel number (property ID) | 5-P Likert Scale |
| Q3.13 | Price of your property | 5-P Likert Scale |
| Q3.14 | Date of property purchase | 5-P Likert Scale |
| Q3.15 | Your car license number | 5-P Likert Scale |
| Q3.16 | Size of your property | 5-P Likert Scale |
| Q3.17 | Your property tax | 5-P Likert Scale |
| Q3.18 | Your floor plan | 5-P Likert Scale |
| Q3.19 | Security Measures (against burglars) on your property | 5-P Likert Scale |
| Q3.20 | Absence of security measures (against burglars) on your property | 5-P Likert Scale |
| Q3.21 | Any other piece of data that may compromise your privacy if shared incidentally? (optional) | 5-P Likert Scale |

Conversely, for 14 of our sub-questions, we chose the 6-point Likert scale. By compelling respondents to lean towards agreement or disagreement, the 6-point scale aids in delineating clearer, more decisive insights into specific attitudes or opinions, which is particularly valuable in areas where a neutral stance is less informative or relevant to our research objectives [32]. The absence of a neutral midpoint in the 6-point scale is instrumental in scenarios where decisiveness in responses is critical, or neutrality could result in ambiguous data interpretation [31,33]. We intended to compel respondents to take a definitive stance on Q7.1–Q7.14, as shown in Table A4, thereby eliminating the central tendency bias, where participants might gravitate towards a neutral choice.

### 3.3. Questions

Our main survey consisted of 14 main questions and 72 sub-questions. For the initial questions Q1 and Q2, as seen in Table 1, participants were presented with two example scenarios, one for each question. Each scenario was made up of three pictures, and we asked if they would share the content publicly if it was theirs. The questions consisted of a 5-P Likert scale entry in combination with an open-ended sub-question. Because such reflective questions can be influenced by later questions in the survey [5], we ask these questions first.

For Q1, participants were presented with Example 1 consisting of three images from a video where someone shows the surroundings of a rural area that can be assumed to be their home, as seen in Figure 2. The video title: Wild Oklahoma Weather, indicates that the video is about an upcoming severe storm. The participants were asked to consider if they would share the video if the depicted house was theirs. For Q2, participants were presented with Example 2, consisting of three social media postings, as seen in Figure 3, and we ask them similarly if they would share the images publicly if they depicted their own property and surroundings.

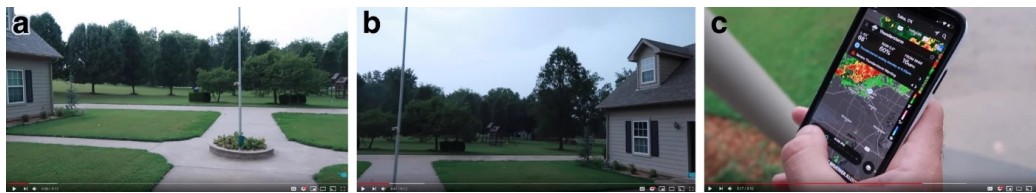

**Figure 2.** Shows the pictures presented to participants in first question Q1 and Q1.1. Subfigures (**a**–**c**) show different scenes from the video. (**a**,**b**) combined hints the shape of the backyard, whereas (**c**) depicts a smartphone with a weather app showing an incoming storm and the current position as a blue dot. The pictures were taken from Kutschera [6] and [34], respectively.

Question Q3, as seen in Table 2, asks the participants about their privacy perceptions on the various data types that are detectable from the social media postings detectable in Example 2 according to the OSINT analysis method proposed by Kutschera [6]. Several other key sensitive data types, like date of birth (Q3.3), blood type (Q3.6), and social security number (Q3.7), are also included.

Questions Q4 and Q5, as seen in Table 3, ask participants about their perception of various privacy guidelines they practice currently (Q4) and in the future (Q5). The questions Q4 and Q5 differ only in their usage, namely, Q4-current and Q5-future. As the guidelines are the same, they are best represented in joined Table 3. The purpose of Q5 is to see to what extent participants had their perceptions influenced by participating in this survey.

Questions Q6 and Q7 are about social media usage (Tables A3 and A4). Questions Q8–Q14, which are about demographic values, as seen in Table A5, were implemented. We use these demographic data to organize respondents into various sub-group filters. The abbreviations on the filters used for each subgroup are listed and explained in Table 4. Besides these active responses, the survey provider also collected the start and end timestamps for each survey. These start and end times allow us to calculate the time spent on the survey.

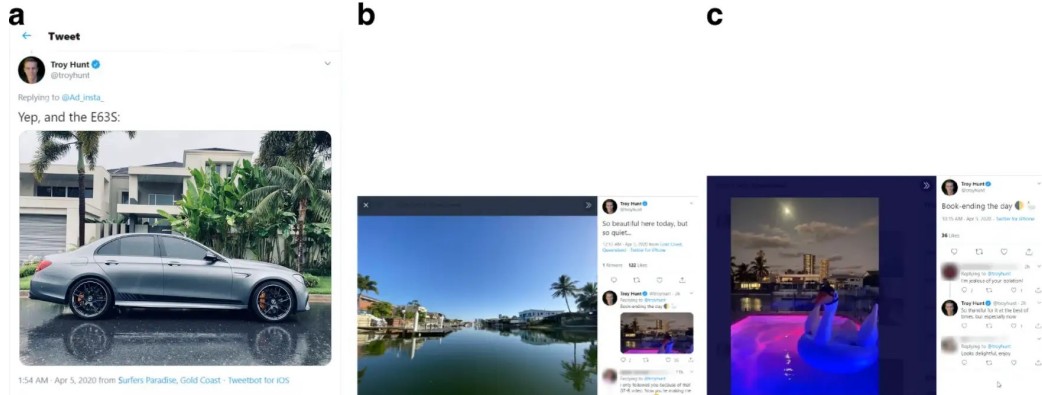

**Figure 3.** Shows the pictures presented to participants in second question Q2 and Q2.1. Each subfigure (**a**–**c**) represents different postings from the same person on Twitter. The posting shown in (**a**) is a response to a question if the car is still owned, (**b**) an untriggered comment about how beautiful the day is, (**c**) depicts a posting that comments on the end of the day where the skyline and a small pool is visible alongside the moon. The pictures were taken from Kutschera [6] and Twitter [35–37], respectively.

**Table 3.** Survey IDS2301 Questions 4; current guideline usage and Questions 5; future guideline usage. The numbering of awareness guidelines is aligned through Q4.n and Q5.n.

| Ref. | | Question | Type |
|---|---|---|---|
| | Q4 | What measures against the accidental posting of private information do you currently use actively when posting on social media | |
| Q5 | | What measures against the accidental posting of private information will you additionally use in the future? (optional) | |
| Q5.0 | | None other than those before. | Boolean |
| Q5.1 | Q4.1 | Avoid posting content that includes house numbers or street names. | Boolean |
| Q5.2 | Q4.2 | Be on the lookout for reflections in mirrors as well as on surfaces such as cars, windows, vitrines, glasses, sunglasses, or watches. | Boolean |
| Q5.3 | Q4.3 | Post content of vacations—if at all—only after the vacation has ended. | Boolean |
| Q5.4 | Q4.4 | Avoid repetition of vacations or periods of absence, such as "during New Years Eve I am—always—on a one-week trip". | Boolean |
| Q5.5 | Q4.5 | Posts should be in accordance with a single time zone irrespective of a current and temporary time zone. | Boolean |
| Q5.6 | Q4.6 | Avoid posting any information from parcels or letters, such as tracking number, full address, names, or QR codes. | Boolean |
| Q5.7 | Q4.7 | Do not post IDs such as driver's license, personal ID, credit or debit card, et cetera. | Boolean |
| Q5.8 | Q4.8 | When posting letters, make sure that the address or sensitive information on the back of the letterhead does not show through. | Boolean |
| Q5.9 | Q4.9 | Avoid posting scenes that include location-based map materials, such as navigation maps, weather or fitness apps, et cetera. | Boolean |
| Q5.10 | Q4.10 | Close all curtains or post content where no windows are visible. | Boolean |
| Q5.11 | Q4.11 | Try tilting the camera angle as low as possible when showing your own property. | Boolean |
| Q5.12 | Q4.12 | Be aware that shadows or the sun's position can also hint at additional information about the location. | Boolean |
| Q5.13 | Q4.13 | Do not share fitness routes that start or end at your home location. | Boolean |
| Q5.14 | Q4.14 | Do not share information about your own or surrounding WLAN/WiFi SSIDs | Boolean |
| | Q4.15 | None of the above | Boolean |
| Q5.15 | Q4.16 | Other | open-ended |

**Table 4.** Lists the abbreviations and their filtered participants of the analysed subsets of the survey. The letters of the abbreviation's origin are underlined within the description.

| Abbreviation | Description |
|---|---|
| ALL | ALL survey participants, unfiltered |
| ANP | All Not likely to Publish example #1 and #2 |
| ALP | All Likely to Publish example #1 and #2 |
| E1P | Example #1 likely to Publish |
| E2P | Example #2 likely to Publish |
| BY | Bachelor's degree Yes |
| BN | Bachelor's degree No |
| LR | Lives in a Rented property |
| LO | Lives in an Owned property |
| AS | Age is Smaller or equal to 24 |
| AB | Age is Bigger than 24 |
| BYS | Bachelor's degree Yes and age Smaller or equal to 24 |
| BNS | Bachelor's degree No and age Smaller or equal to 24 |
| LRS | Lives in a Rented property and age Smaller or equal to 24 |
| LOS | Lives in an Owned property and age Smaller or equal to 24 |
| BYB | Bachelor's degree Yes and age Bigger than 24 |
| BNB | Bachelor's degree No and age Bigger than 24 |
| LRB | Lives in a Rented property and age Bigger than 24 |
| LOB | Lives in an Owned property and age Bigger than 24 |
| S250 | Lives in city with population Smaller than 250,000 people |
| B250 | Lives in city with population Bigger than 250,000 people |
| SCC | Social media filter, Consume Content response greater or equal to 4. |
| SUC | Social media filter, Uploads Content response greater or equal to 4. |
| M | Male |
| F | Female |
| SD | Standard Deviation |
| MOE | Margin of Error |

## 4. Results

Table 5 shows the percentage of participants who responded with either agree or strongly agree on questions about the privacy compromise for the various data types listed in questions Q3.1–Q3.20. The percentage of those that disagreed or strongly disagreed is shown in Table A6. The background color in both tables is graded from green to red through yellow based on the cell value. Within both tables, the data type that can be found is shown more visually within rows E.1 and E.2, respectively. Further, the data types correlate to questions Q3.1–Q3.20.

The boxplot in Figure 4 depicts the statistical properties of the responses, with the median at the tapered point with an orange line and supports the results presented in Table 5. The adjoining areas indicate the 25% above and below the median, the whiskers indicate the first and fourth quartile of responses, and outliers are indicated by a circle.

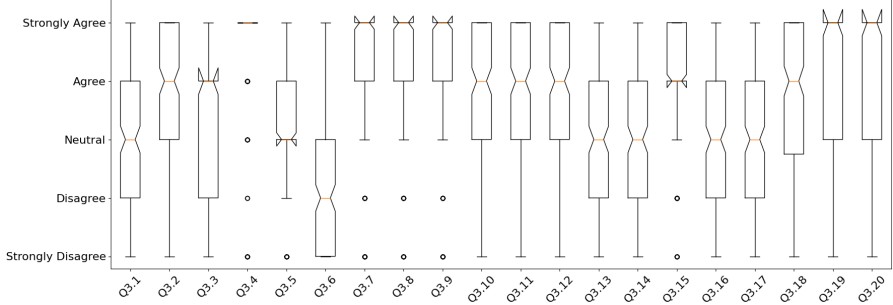

**Figure 4.** Depicts the boxplot visualizing the statistical values, such as median and the quantiles of all answers, from Q3.1 to Q3.20.

**Table 5.** Shows the data types from Question 3 (Q3.1–Q3.20). In rows *Example 1* (E.1) and *Example 2* (E.2) the data types found are marked with *Yes (X)* and *No (O)*. Column N states the number of participants in each row. Column *Abbr.* lists all subgroups using their filter identifier as explained in Table 4. The cells represent the percentage of participants of the said subgroup who responded with either *agree* **or** *strongly agree* on questions about the privacy compromise regarding data types Q3.1–Q3.20. Based on the cell value, the background color is graded from green to red through yellow.

| N | Abbr. | Q3.n | | | | | | | | | | | | | | | | | | | |
| | | 1 | 2 | 3 | 4 | 5 | 6 | 7 | 8 | 9 | 10 | 11 | 12 | 13 | 14 | 15 | 16 | 17 | 18 | 19 | 20 |
|---|---|---|---|---|---|---|---|---|---|---|---|---|---|---|---|---|---|---|---|---|---|
| | E.1 | X | X | O | X | X | O | O | X | O | O | O | X | X | X | O | X | X | O | X | X |
| | E.2 | X | X | O | X | O | O | O | X | X | O | O | X | X | X | O | X | O | O | X | X |
| 192 | ALL | 33.33 | 74.48 | 51.04 | 87.50 | 47.92 | 21.35 | 80.73 | 77.08 | 83.33 | 57.81 | 64.58 | 67.71 | 33.85 | 34.38 | 76.04 | 29.69 | 35.42 | 53.65 | 71.88 | 73.96 |
| 16 | ALP | 25.00 | 81.25 | 50.00 | 81.25 | 25.00 | 12.50 | 81.25 | 87.50 | 81.25 | 50.00 | 50.00 | 56.25 | 31.25 | 18.75 | 75.00 | 25.00 | 37.50 | 31.25 | 50.00 | 50.00 |
| 88 | ANP | 37.50 | 79.55 | 56.82 | 90.91 | 54.55 | 22.73 | 80.68 | 77.27 | 82.95 | 60.23 | 67.05 | 69.32 | 45.45 | 54.55 | 80.68 | 42.05 | 37.50 | 64.77 | 78.41 | 76.14 |
| 27 | E1P | 37.04 | 77.78 | 44.44 | 81.48 | 37.04 | 18.52 | 77.78 | 77.78 | 77.78 | 51.85 | 62.96 | 62.96 | 33.33 | 14.81 | 77.78 | 33.33 | 48.15 | 37.04 | 59.26 | 66.67 |
| 42 | E2P | 28.57 | 73.81 | 50.00 | 85.71 | 38.10 | 21.43 | 85.71 | 83.33 | 85.71 | 61.90 | 61.90 | 64.29 | 28.57 | 28.57 | 78.57 | 14.29 | 35.71 | 38.10 | 69.05 | 73.81 |
| 73 | BY | 31.51 | 72.60 | 53.42 | 87.67 | 46.58 | 24.66 | 82.19 | 82.19 | 87.67 | 57.53 | 61.64 | 68.49 | 34.25 | 34.25 | 76.71 | 34.25 | 39.73 | 47.95 | 71.23 | 72.60 |
| 119 | BN | 34.45 | 75.63 | 49.58 | 87.39 | 48.74 | 19.33 | 79.83 | 73.95 | 80.67 | 57.98 | 66.39 | 67.23 | 33.61 | 34.45 | 75.63 | 26.89 | 32.77 | 57.14 | 72.27 | 74.79 |
| 136 | LR | 33.82 | 72.79 | 49.26 | 86.76 | 41.18 | 24.26 | 80.15 | 75.00 | 81.62 | 53.68 | 62.50 | 64.71 | 32.35 | 35.29 | 72.06 | 29.41 | 33.82 | 48.53 | 74.26 | 72.06 |
| 50 | LO | 34.00 | 80.00 | 52.00 | 88.00 | 60.00 | 14.00 | 80.00 | 80.00 | 86.00 | 64.00 | 66.00 | 74.00 | 38.00 | 32.00 | 84.00 | 32.00 | 38.00 | 64.00 | 66.00 | 76.00 |
| 119 | AS | 31.93 | 74.79 | 47.06 | 84.03 | 47.06 | 21.01 | 80.67 | 75.63 | 84.03 | 55.46 | 66.39 | 65.55 | 31.09 | 36.13 | 70.59 | 27.73 | 33.61 | 52.10 | 70.59 | 69.75 |
| 67 | AB | 35.82 | 74.63 | 58.21 | 94.03 | 49.25 | 22.39 | 82.09 | 80.60 | 83.58 | 61.19 | 62.69 | 71.64 | 40.30 | 31.34 | 85.07 | 32.84 | 40.30 | 56.72 | 77.61 | 83.58 |
| 29 | BYS | 34.48 | 72.41 | 41.38 | 79.31 | 41.38 | 27.59 | 82.76 | 82.76 | 89.66 | 51.72 | 65.52 | 65.52 | 27.59 | 41.38 | 68.97 | 31.03 | 34.48 | 41.38 | 68.97 | 68.97 |
| 90 | BNS | 31.11 | 75.56 | 48.89 | 85.56 | 48.89 | 18.89 | 80.00 | 73.33 | 82.22 | 56.67 | 66.67 | 65.56 | 32.22 | 34.44 | 71.11 | 26.67 | 33.33 | 55.56 | 71.11 | 70.00 |
| 86 | LRS | 32.56 | 72.09 | 46.51 | 82.56 | 40.70 | 25.58 | 81.40 | 72.09 | 82.56 | 52.33 | 62.79 | 62.79 | 27.91 | 38.37 | 66.28 | 29.07 | 34.88 | 47.67 | 69.77 | 65.12 |
| 32 | LOS | 31.25 | 81.25 | 46.88 | 87.50 | 62.50 | 9.38 | 78.12 | 84.38 | 87.50 | 62.50 | 75.00 | 71.88 | 37.50 | 31.25 | 81.25 | 25.00 | 28.12 | 62.50 | 71.88 | 81.25 |
| 41 | BYB | 29.27 | 73.17 | 65.85 | 95.12 | 53.66 | 24.39 | 85.37 | 85.37 | 90.24 | 63.41 | 63.41 | 73.17 | 41.46 | 31.71 | 82.93 | 36.59 | 46.34 | 56.10 | 78.05 | 80.49 |
| 26 | BNB | 46.15 | 76.92 | 46.15 | 92.31 | 42.31 | 19.23 | 76.92 | 73.08 | 73.08 | 57.69 | 61.54 | 69.23 | 38.46 | 30.77 | 88.46 | 26.92 | 30.77 | 57.69 | 76.92 | 88.46 |
| 49 | LRB | 34.69 | 73.47 | 55.10 | 93.88 | 42.86 | 22.45 | 79.59 | 81.63 | 81.63 | 57.14 | 63.27 | 69.39 | 40.82 | 30.61 | 81.63 | 28.57 | 32.65 | 51.02 | 83.67 | 85.71 |
| 16 | LOB | 43.75 | 81.25 | 68.75 | 93.75 | 62.50 | 25.00 | 87.50 | 75.00 | 87.50 | 68.75 | 56.25 | 81.25 | 43.75 | 37.50 | 93.75 | 50.00 | 62.50 | 75.00 | 62.50 | 75.00 |
| 52 | S250 | 32.69 | 75.00 | 46.15 | 90.38 | 44.23 | 9.62 | 80.77 | 84.62 | 90.38 | 61.54 | 69.23 | 69.23 | 42.31 | 38.46 | 73.08 | 32.69 | 28.85 | 51.92 | 67.31 | 75.00 |
| 134 | B250 | 33.58 | 74.63 | 52.24 | 85.82 | 47.76 | 26.12 | 80.60 | 73.88 | 80.60 | 55.22 | 61.94 | 66.42 | 30.60 | 32.84 | 76.12 | 28.36 | 38.06 | 53.73 | 73.88 | 73.13 |
| 161 | SCC | 31.06 | 74.53 | 50.93 | 86.96 | 46.58 | 21.74 | 80.75 | 77.64 | 83.23 | 56.52 | 66.46 | 67.70 | 30.43 | 30.43 | 72.67 | 27.95 | 32.92 | 50.31 | 69.57 | 72.67 |
| 21 | SUC | 47.62 | 52.38 | 47.62 | 66.67 | 33.33 | 23.81 | 66.67 | 71.43 | 71.43 | 57.14 | 57.14 | 42.86 | 28.57 | 33.33 | 71.43 | 42.86 | 42.86 | 52.38 | 57.14 | 57.14 |
| 156 | M | 35.90 | 75.64 | 48.72 | 86.54 | 48.08 | 19.87 | 78.21 | 73.08 | 82.05 | 56.41 | 61.54 | 66.67 | 33.33 | 32.69 | 72.44 | 29.49 | 33.33 | 51.92 | 69.23 | 72.44 |
| 32 | F | 18.75 | 65.62 | 56.25 | 90.62 | 43.75 | 31.25 | 90.62 | 93.75 | 87.50 | 62.50 | 75.00 | 71.88 | 34.38 | 43.75 | 90.62 | 28.12 | 40.62 | 59.38 | 84.38 | 81.25 |
| 192 | SD | 1.26 | 1.16 | 1.23 | 1.09 | 1.23 | 1.32 | 1.40 | 1.27 | 1.22 | 1.25 | 1.23 | 1.32 | 1.21 | 1.23 | 1.17 | 1.22 | 1.30 | 1.35 | 1.31 | 1.30 |
| 192 | MOE | 2.47 | 2.28 | 2.41 | 2.14 | 2.42 | 2.59 | 2.74 | 2.48 | 2.39 | 2.44 | 2.41 | 2.59 | 2.37 | 2.42 | 2.29 | 2.39 | 2.55 | 2.65 | 2.57 | 2.55 |

Row E.1 within Table 5 marks the data types extractable from Example 1, as shown in Figure 2. Namely, the state or country (Q3.1), full name (Q3.2), full address (Q3.4), full name of previous owners (Q3.5), information on relatives (Q3.8), parcel number of the property (Q3.12), price of the property (Q3.13), date of purchase of property (Q3.14), size of the property (Q3.16), property tax (Q3.17), and security measures against burglars (Q3.19) as well as the absence thereof (Q3.20).

Row E.2 within Table 5 marks extractable from Example 2, as shown in Figure 3. Namely, the state or country (Q3.1), full name (Q3.2), full address (Q3.4), information on relatives (Q3.8), phone number (Q3.9), parcel number of the property (Q3.12), price of the property (Q3.13), date of purchase of property (Q3.14), size of the property (Q3.16), and security measures against burglars (Q3.19) as well as the absence thereof (Q3.20).

The percentage of positive answers to Q4 towards the current usage of awareness guidelines are shown in Table 6, while the percentage of positive answers to Q5 towards the future usage of awareness guidelines are shown in Table 7. The same filter groups are used as for Q3 in Table 5. Based on the cell value, the background color is graded from green to red through yellow.

In Q6 and Q7 of our survey, participants were asked to optionally answer questions about their social media usage, what social media platforms they use, and to what extent on a 6P-Likert Scale with the following options: no answer (0), never (1), very rarely (2), rarely (3), occasionally (4), frequently (5), and very frequently (6), with 0 as the default value. The results are visualized in Table A2, whereas the questions are listed in Tables A3 and A4. The background color in Table A2 is determined by the value of the cell from green to red through yellow.

**Table 6.** Shows the percentage of positive answers towards current usage of guidelines from Q4. Based on the cell value, the background color is graded from green to red through yellow.

| N | Abbr. | Q4.n | | | | | | | | | | | | | | |
|---|---|---|---|---|---|---|---|---|---|---|---|---|---|---|---|---|
| | | 1 | 2 | 3 | 4 | 5 | 6 | 7 | 8 | 9 | 10 | 11 | 12 | 13 | 14 | 15 |
| 192 | ALL | 78.65 | 42.71 | 36.98 | 33.85 | 11.46 | 80.21 | 86.46 | 72.40 | 46.35 | 15.10 | 19.79 | 8.33 | 44.27 | 65.10 | 3.65 |
| 16 | ALP | 62.50 | 18.75 | 12.50 | 12.50 | 6.25 | 75.00 | 87.50 | 62.50 | 25.00 | 6.25 | 6.25 | 0.00 | 12.50 | 56.25 | 6.25 |
| 88 | ANP | 82.95 | 43.18 | 43.18 | 37.50 | 10.23 | 78.41 | 81.82 | 65.91 | 48.86 | 21.59 | 26.14 | 9.09 | 54.55 | 63.64 | 3.41 |
| 27 | E1P | 70.37 | 37.04 | 25.93 | 25.93 | 14.81 | 70.37 | 81.48 | 59.26 | 25.93 | 14.81 | 14.81 | 7.41 | 22.22 | 59.26 | 7.41 |
| 42 | E2P | 66.67 | 26.19 | 26.19 | 16.67 | 4.76 | 80.95 | 92.86 | 80.95 | 38.10 | 4.76 | 7.14 | 2.38 | 30.95 | 69.05 | 2.38 |
| 73 | BY | 75.34 | 41.10 | 30.14 | 30.14 | 10.96 | 86.30 | 87.67 | 72.60 | 35.62 | 13.70 | 17.81 | 9.59 | 43.84 | 65.75 | 2.74 |
| 119 | BN | 80.67 | 43.70 | 41.18 | 36.13 | 11.76 | 76.47 | 85.71 | 72.27 | 52.94 | 15.97 | 21.01 | 7.56 | 44.54 | 64.71 | 4.20 |
| 136 | LR | 73.53 | 39.71 | 36.03 | 32.35 | 13.24 | 77.21 | 84.56 | 69.85 | 43.38 | 15.44 | 22.06 | 8.82 | 40.44 | 60.29 | 5.15 |
| 50 | LO | 90.00 | 52.00 | 36.00 | 36.00 | 6.00 | 86.00 | 90.00 | 78.00 | 54.00 | 14.00 | 14.00 | 8.00 | 54.00 | 76.00 | 0.00 |
| 119 | AS | 79.83 | 45.38 | 37.82 | 33.61 | 12.61 | 77.31 | 85.71 | 71.43 | 44.54 | 15.97 | 19.33 | 9.24 | 42.86 | 62.18 | 3.36 |
| 67 | AB | 74.63 | 37.31 | 35.82 | 34.33 | 10.45 | 83.58 | 86.57 | 71.64 | 47.76 | 11.94 | 19.40 | 7.46 | 46.27 | 68.66 | 4.48 |
| 29 | BYS | 68.97 | 44.83 | 34.48 | 27.59 | 10.34 | 79.31 | 82.76 | 68.97 | 20.69 | 17.24 | 10.34 | 10.34 | 34.48 | 48.28 | 3.45 |
| 90 | BNS | 83.33 | 45.56 | 38.89 | 35.56 | 13.33 | 76.67 | 86.67 | 72.22 | 52.22 | 15.56 | 22.22 | 8.89 | 45.56 | 66.67 | 3.33 |
| 86 | LRS | 75.58 | 40.70 | 36.05 | 30.23 | 13.95 | 74.42 | 82.56 | 68.60 | 40.70 | 17.44 | 23.26 | 9.30 | 37.21 | 55.81 | 4.65 |
| 32 | LOS | 90.62 | 56.25 | 40.62 | 40.62 | 9.38 | 84.38 | 93.75 | 78.12 | 53.12 | 12.50 | 9.38 | 9.38 | 56.25 | 78.12 | 0.00 |
| 41 | BYB | 78.05 | 36.59 | 29.27 | 31.71 | 12.20 | 90.24 | 90.24 | 73.17 | 43.90 | 9.76 | 21.95 | 9.76 | 48.78 | 75.61 | 2.44 |
| 26 | BNB | 69.23 | 38.46 | 46.15 | 38.46 | 7.69 | 73.08 | 80.77 | 69.23 | 53.85 | 15.38 | 15.38 | 3.85 | 42.31 | 57.69 | 7.69 |
| 49 | LRB | 69.39 | 38.78 | 36.73 | 34.69 | 12.24 | 81.63 | 87.76 | 71.43 | 46.94 | 12.24 | 20.41 | 8.16 | 44.90 | 67.35 | 6.12 |
| 16 | LOB | 87.50 | 37.50 | 31.25 | 31.25 | 0.00 | 87.50 | 81.25 | 75.00 | 56.25 | 12.50 | 18.75 | 6.25 | 50.00 | 68.75 | 0.00 |
| 52 | S250 | 78.85 | 42.31 | 36.54 | 30.77 | 11.54 | 73.08 | 80.77 | 73.08 | 42.31 | 7.69 | 13.46 | 3.85 | 38.46 | 63.46 | 1.92 |
| 134 | B250 | 77.61 | 43.28 | 36.57 | 34.33 | 11.94 | 82.09 | 88.06 | 70.90 | 47.01 | 17.91 | 22.39 | 10.45 | 45.52 | 64.93 | 4.48 |
| 161 | SCC | 80.75 | 44.10 | 34.78 | 32.92 | 10.56 | 81.99 | 90.06 | 75.78 | 47.20 | 14.91 | 18.63 | 5.59 | 43.48 | 65.84 | 3.73 |
| 21 | SUC | 80.95 | 28.57 | 19.05 | 14.29 | 9.52 | 80.95 | 90.48 | 90.48 | 33.33 | 9.52 | 9.52 | 4.76 | 61.90 | 66.67 | 4.76 |
| 156 | M | 80.77 | 46.79 | 38.46 | 35.26 | 13.46 | 78.21 | 84.62 | 69.23 | 46.79 | 16.03 | 21.15 | 10.26 | 44.87 | 67.95 | 4.49 |
| 32 | F | 68.75 | 18.75 | 28.12 | 28.12 | 3.12 | 90.62 | 96.88 | 87.50 | 40.62 | 9.38 | 15.62 | 0.00 | 43.75 | 56.25 | 0.00 |

**Table 7.** Shows the percentage of positive answers towards future usage of guidelines from Q5.

| N | Abbr. | 0 | 1 | 2 | 3 | 4 | 5 | 6 | 7 | 8 | 9 | 10 | 11 | 12 | 13 | 14 |
|---|---|---|---|---|---|---|---|---|---|---|---|---|---|---|---|---|
| 192 | ALL | 42.71 | 9.90 | 16.15 | 14.58 | 11.46 | 6.77 | 10.94 | 11.46 | 11.46 | 14.06 | 5.21 | 13.02 | 11.46 | 14.06 | 12.50 |
| 16 | ALP | 62.50 | 12.50 | 6.25 | 6.25 | 6.25 | 0.00 | 12.50 | 12.50 | 12.50 | 6.25 | 0.00 | 0.00 | 0.00 | 12.50 | 6.25 |
| 88 | ANP | 31.82 | 7.95 | 13.64 | 18.18 | 11.36 | 9.09 | 10.23 | 12.50 | 10.23 | 17.05 | 6.82 | 14.77 | 14.77 | 14.77 | 15.91 |
| 27 | E1P | 55.56 | 14.81 | 11.11 | 11.11 | 11.11 | 7.41 | 14.81 | 14.81 | 14.81 | 11.11 | 11.11 | 14.81 | 11.11 | 14.81 | 7.41 |
| 42 | E2P | 52.38 | 9.52 | 19.05 | 7.14 | 11.90 | 2.38 | 11.90 | 11.90 | 11.90 | 9.52 | 2.38 | 9.52 | 11.90 | 16.67 | 9.52 |
| 73 | BY | 45.21 | 8.22 | 15.07 | 9.59 | 13.70 | 2.74 | 10.96 | 12.33 | 13.70 | 16.44 | 4.11 | 10.96 | 8.22 | 9.59 | 13.70 |
| 119 | BN | 41.18 | 10.92 | 16.81 | 17.65 | 10.08 | 9.24 | 10.92 | 10.92 | 10.08 | 12.61 | 5.88 | 14.29 | 13.45 | 16.81 | 11.76 |
| 136 | LR | 41.91 | 11.03 | 17.65 | 13.97 | 12.50 | 7.35 | 11.76 | 11.76 | 12.50 | 15.44 | 4.41 | 13.24 | 11.76 | 14.71 | 13.24 |
| 50 | LO | 44.00 | 6.00 | 10.00 | 16.00 | 8.00 | 4.00 | 8.00 | 10.00 | 8.00 | 10.00 | 6.00 | 12.00 | 10.00 | 12.00 | 10.00 |
| 119 | AS | 40.34 | 10.92 | 15.13 | 16.81 | 12.61 | 7.56 | 13.45 | 12.61 | 13.45 | 15.97 | 5.88 | 15.13 | 9.24 | 16.81 | 15.13 |
| 67 | AB | 46.27 | 7.46 | 16.42 | 10.45 | 8.96 | 4.48 | 5.97 | 8.96 | 7.46 | 10.45 | 2.99 | 8.96 | 14.93 | 8.96 | 7.46 |
| 29 | BYS | 41.38 | 10.34 | 17.24 | 10.34 | 20.69 | 6.90 | 20.69 | 20.69 | 27.59 | 27.59 | 6.90 | 17.24 | 6.90 | 13.79 | 20.69 |
| 90 | BNS | 40.00 | 11.11 | 14.44 | 18.89 | 10.00 | 7.78 | 11.11 | 10.00 | 8.89 | 12.22 | 5.56 | 14.44 | 10.00 | 17.78 | 13.33 |
| 86 | LRS | 39.53 | 11.63 | 16.28 | 16.28 | 12.79 | 8.14 | 13.95 | 12.79 | 13.95 | 17.44 | 5.81 | 15.12 | 9.30 | 16.28 | 15.12 |
| 32 | LOS | 43.75 | 9.38 | 12.50 | 18.75 | 12.50 | 6.25 | 12.50 | 12.50 | 12.50 | 12.50 | 6.25 | 15.62 | 9.38 | 18.75 | 15.62 |
| 41 | BYB | 46.34 | 7.32 | 14.63 | 9.76 | 9.76 | 0.00 | 4.88 | 7.32 | 4.88 | 9.76 | 2.44 | 7.32 | 9.76 | 7.32 | 9.76 |
| 26 | BNB | 46.15 | 7.69 | 19.23 | 11.54 | 7.69 | 11.54 | 7.69 | 11.54 | 11.54 | 11.54 | 3.85 | 11.54 | 23.08 | 11.54 | 3.85 |
| 49 | LRB | 46.94 | 10.20 | 20.41 | 10.20 | 12.24 | 6.12 | 8.16 | 10.20 | 10.20 | 12.24 | 2.04 | 10.20 | 16.33 | 12.24 | 10.20 |
| 16 | LOB | 37.50 | 0.00 | 6.25 | 12.50 | 0.00 | 0.00 | 0.00 | 6.25 | 0.00 | 6.25 | 6.25 | 6.25 | 12.50 | 0.00 | 0.00 |
| 52 | S250 | 32.69 | 11.54 | 17.31 | 21.15 | 13.46 | 9.62 | 11.54 | 13.46 | 9.62 | 11.54 | 5.77 | 15.38 | 9.62 | 17.31 | 11.54 |
| 134 | B250 | 47.01 | 8.96 | 14.93 | 11.94 | 10.45 | 5.22 | 10.45 | 10.45 | 11.94 | 14.93 | 4.48 | 11.94 | 11.94 | 12.69 | 12.69 |
| 161 | SCC | 44.10 | 9.94 | 15.53 | 14.29 | 12.42 | 5.59 | 11.18 | 10.56 | 11.18 | 13.04 | 5.59 | 13.66 | 12.42 | 15.53 | 13.04 |
| 21 | SUC | 33.33 | 4.76 | 14.29 | 9.52 | 19.05 | 0.00 | 9.52 | 4.76 | 14.29 | 28.57 | 4.76 | 9.52 | 9.52 | 4.76 | 9.52 |
| 156 | M | 45.51 | 7.69 | 13.46 | 13.46 | 7.69 | 5.77 | 8.97 | 8.97 | 7.69 | 10.26 | 5.13 | 13.46 | 10.26 | 11.54 | 9.62 |
| 32 | F | 31.25 | 18.75 | 28.12 | 18.75 | 28.12 | 9.38 | 18.75 | 21.88 | 28.12 | 31.25 | 3.12 | 9.38 | 15.62 | 25.00 | 25.00 |

The privacy awareness guidelines proposed by Kutschera [6] are enumerated in Table 3. Table 8 shows how each guideline can prevent the exposure of a certain data type. For instance, enforcing guideline Q4.4 will help minimize the exposure risk of current state or country (Q3.1), date of birth (Q3.3), security measures against burglars (Q3.19), and absence of security measures against burglars (Q3.20). Naturally, OSINT has manifold ways of detecting data types, and some can be obtained by gaining knowledge of another data type first. For example, price of property (Q3.13) or date of property purchase (Q3.14) may become evident through the detection of full address (Q3.4). Those data types are listed in the Indirect column of Table 8.

**Table 8.** Mapping of data types to guidelines.

| Direct | Indirect | 1 | 2 | 3 | 4 | 5 | 6 | 7 | 8 | 9 | 10 | 11 | 12 | 13 | 14 |
|---|---|---|---|---|---|---|---|---|---|---|---|---|---|---|---|---|
| Q3.1 | | X | X | X | X | X | X | X | X | X | X | X | X | X | X |
| Q3.2 | | | | | | | X | X | X | | | | | | |
| Q3.3 | | | | | X | | X | X | X | X | X | X | X | X | X |
| Q3.4 | | X | X | | | X | X | | X | X | X | X | X | X | |
| Q3.5 | (Q3.4) | X | X | | | | X | | X | | | | | | |
| Q3.6 | | | | | | | | | X | | | | | | |
| Q3.7 | | | | | | | | | X | | | | | | |
| Q3.8 | (Q3.2) | | | | | | X | X | X | | | | | | |
| Q3.9 | (Q3.2) | | | | | | | | X | | | | | | |
| Q3.10 | (Q3.2) | | | | | | | | X | | | | | | |
| Q3.11 | | | | X | | | X | | X | | | | | | |
| Q3.12 | | X | X | | | | X | | X | | X | X | X | X | X |
| Q3.13 | (Q3.4) | X | X | | | | X | | X | X | X | X | X | X | X |
| Q3.14 | (Q3.4) | X | X | | | | X | | X | X | X | X | X | X | X |
| Q3.15 | | | | X | | | | | X | | | | X | X | |
| Q3.16 | (Q3.4) | X | X | | | | X | | X | X | X | X | X | X | X |
| Q3.17 | (Q3.4) | X | X | | | | X | | X | X | X | X | X | X | X |
| Q3.18 | (Q3.4) | X | X | | | | X | | X | X | X | X | X | X | X |
| Q3.19 | | | X | X | X | | X | | X | | X | X | X | | |
| Q3.20 | | | X | X | X | | | | | | | X | X | | |

## 5. Discussions

We found that a two-thirds supermajority of the participants have privacy concerns about data types Q3.2, Q3.4, Q3.7, Q3.8, Q3.9, Q3.12, Q3.15, Q3.19, and Q3.20. For each of these data types, Table 9 shows the percentage of respondents in various subgroups who agree or strongly agree to post either or both Example 1 and Example 2, matched with the privacy concerns of the group. Rows E.1 and E.2 indicate whether or not the data type can actually be found in the examples. Cell color corresponding to the filtered groups (ALP, E1P, and E2P) illustrates the distribution of majority levels in cases where the data type can be revealed. Cells that reach a simple majority (50%) are highlighted in yellow, while the ones reaching a two-thirds majority (66.67%) are highlighted in orange.

**Table 9.** Shows excerpt of data types from Question 3 (Q3.n) (Table 5). It displays the presence (X) or absence (O) of specific data types in 'Example 1' (E.1) and 'Example 2' (E.2). The 'Abbr.' column lists subgroups using filter identifiers (see Table 4). Cells indicate the percentage of subgroup participants agreeing or strongly agreeing to publish the posting. Cell highlighting denotes extractable data types with a simple majority in yellow and a supermajority in orange.

| N | N% | Abbr. ** | Q3.n * | | | | | | | | |
|---|----|----------|------|------|------|------|------|------|------|------|------|
| | | | 2 | 4 | 7 | 8 | 9 | 12 | 15 | 19 | 20 |
| | | E.1 | X | X | O | X | O | X | O | X | X |
| | | E.2 | X | X | O | X | X | X | O | X | X |
| 192 | 100.00 | ALL | 74.48 | 87.5 | 80.73 | 77.08 | 83.33 | 67.71 | 76.04 | 71.88 | 73.96 |
| 16 | 8.33 | ALP | 81.25 | 81.25 | 81.25 | 87.5 | 81.25 | 56.25 | 75.0 | 50.0 | 50.0 |
| 27 | 14.06 | E1P | 77.78 | 81.48 | 77.78 | 77.78 | 77.78 | 62.96 | 77.78 | 59.26 | 66.67 |
| 42 | 21.88 | E2P | 73.81 | 85.71 | 85.71 | 83.33 | 85.71 | 64.29 | 78.57 | 69.05 | 73.81 |

* Q3.2—"Full name"; Q3.4—"Full address"; Q3.7—"Social security number"; Q3.8—"Information on relatives"; Q3.9—"Phone number"; Q3.12—"Parcel number (property ID)"; Q3.15—"Car license number"; Q3.19—"Security measures"; Q3.20—"Absence of security measures". ** ALL—"ALL survey participants, unfiltered"; ALP—"All Likely to Publish example #1 and #2"; E1P—"Example #1 likely to Publish"; E2P—"Example #2 likely to Publish".

### 5.1. Evaluation and Interpretation of Survey Results

Our study aims to detect privacy awareness on social media implicitly. Alongside the methodology used, this study never asked or measured direct awareness about incidental data as a direct question, as this might have influenced the participant and thus rendered this study invalid. Moreover, we used well-analyzed postings from our previous research, of which we knew exactly what data types could be discovered in a strict time frame of up to two hours. In the first step, the participants had to answer whether they would have posted the content shown, see Table 1. In the second step, the participants had to answer which data type would compromise their privacy if shared, see Table 2. By combining the results of both questions and the data types found in each example, we gained implicit knowledge about whether the participants would have shared a certain data type and also had concerns about this data type, as shown in Table 5. Below, we split the evaluation into topical sections to evaluate and interpret the present survey results from this study.

#### 5.1.1. Implicit Incidental Data Awareness

Upon taking a closer look, it becomes evident that in certain cases, a supermajority of people concerned about their privacy with regard to a specific data type are willing to share content that can be used to reveal those specific data types.

For example, according to the definition, all 27 individuals in subgroup E1P would publish Example 1 as seen in Figure 2. This example includes, among others, extractable data types full name (Q3.2), full address (Q3.4), information on relatives (Q3.8), and absence of security measures (Q3.20). Although, more than two-thirds of subgroup E1P express concern about these same data types as follows: Q3.2 (77.78%), Q3.4 (81.48%), Q3.8 (77.78%), and Q3.20 (66.67%).

Furthermore, more than two-thirds of subgroup E2P, who are likely to publish corresponding posting as shown in Figure 3, are concerned about the following data types Q3.2 (73.81%), Q3.4 (85.71%), Q3.8 (83.33%), Q3.9 (85.71%), Q3.19 (69.05%), and Q3.20 (73.81%).

Data type parcel number (Q3.12) was included, but it did only reach a single majority of 62.96% (E1P) and 64.29% (E2P), respectively. The discussed details are visible in Table 9, which is an excerpt of Table 5.

In summary, the participants of E1P and E2P are concerned about data types Q3.2, Q3.4, Q3.8, and Q3.20, but are also very likely to share a post containing those data types. This allows us to draw an implicit conclusion that these individuals are unaware of incidental data contained in certain postings. Together these results provide important insights into the awareness of sharing privacy concerning incidental data.

### 5.1.2. Notable Results from Opposite Filter Groups

Overall, the privacy concern in group ALL with regard to data types Q3.2 (74.48%), Q3.4 (87.5%), Q3.8 (77.08%), and Q3.20 (73.96%) is high, as Table 5 shows. At the same time, a look at Table A6 reveals that between 18.75% and 6.25% disagree that the data types Q3.2 (18.75%), Q3.4 (6.25%), Q3.8 (14.58%), and Q3.20 (17.19%) are privacy-compromising, which further confirms our findings.

Furthermore, interesting is that 93.75% of those in subgroup LOB are concerned about the car license number (Q3.15), but only 76.04% of the overall group ALL and 66.28% of subgroup LRS, respectively, are concerned about the same data type. The reason for this could either be that people who lived on their own property are more aware of what can be revealed or what harm can be performed through a car license number, or the meaning of car license number was misunderstood for something other than a license plate.

As for data type full name of previous owners (Q3.5), 62.5% of those in the LOB and LOS groups are concerned, whereas in the ALL group 47.92% are concerned. Even significantly lower is the concern in subgroup ALP with 25.0%, and 40.7% for the LRS subgroup. An indication of a decrease in concern could be that property owners are more aware of potential risks that the name of previous owners can pose in comparison with people who live in rented accommodation.

Another subgroup of interest is LRB where 83.67% are concerned about Q3.19 (security measures against burglars) but only 62.5% of LOB are concerned whereas in the overall group ALL 71.88% are concerned. A possible explanation for this is that people who live on their own property have full power over installation and can also choose on their own to implement concealed and potentially strong measures against burglars, whereas people who live in rented accommodation need the approval of the landlord and will not get compensated in case they move to different housing. This reasoning could lead to a decision for a cheaper movable, and thus non-concealed measures against burglars.

The results that surprised us the most were that subgroup SUC has fewer concerns in each of the most concerned data types. Moreover, compared with group ALL, the concern is lowered on each data type except for current state or country (Q3.1) 47.62%, blood type (Q3.6) 23.81%, size of property (Q3.16) 42.86%, and property tax (Q3.17) 42.86%. A supermajority within the group SUC is concerned about full address (Q3.4) 66.67%, social security number (Q3.7) 66.67%, information on relatives (Q3.8) 71.43%, phone number (Q3.9) 71.43%, and the car license number (Q3.15) 71.43%.

### 5.1.3. Usage of Guidelines

From Table 6, we observe that a two-thirds supermajority of participants (i.e., subgroup ALL) currently use guidelines Q4.1, Q4.6, Q4.7, and Q4.8, whereas Q4.5, Q4.10, Q4.11, and Q4.12 are currently only used by one-third of the same group. In contrast, the answers regarding future usage of the guidelines, Table 7 shows the highest response on Q5.2 (Avoiding reflections on surfaces and mirrors), and the least response with regard to future usage on Q5.10 (Close curtains or avoid windows). The most feedback received was on Q5.0 (None other than those before) with 42.71%. Merely asking questions about privacy

concerns is influencing participants [5]. These results and the overall low rate of response on future usage of the mentioned guidelines suggest that the survey design did not greatly influence the participants.

From subgroup ALP's responses on Q4.9–Q4.11 and Q5.9–Q5.11, as seen in Table 3, we see that few countermeasures are in place today and that this situation will likely be the same in the future. This is interesting as Table 5 shows that 81.25% of the same subgroup ALP "Agree" and "Strongly Agree" that data type full address (Q3.4) can compromise their privacy. Moreover, it can be assumed that full address (Q3.4) can be discovered very quickly when map material (Q4.9 and Q5.9) is included in a post, which the subgroup ALP would publish as per the definition in Table 4.

As discussed in Section 5.1.1, subgroups E1P and E2P are willing to post data that a majority of the group has privacy concerns about. Furthermore, the usage of the guidelines in Table 8 reveals that the measures stated in the guidelines, which may well have prevented the publication of incidental data, are not used. For example, in order to avoid data type Q3.4 incidental data, one can focus on guidelines Q4.1, Q4.2, Q4.5, Q4.6, and Q4.8–13. Only two to three guidelines are used by a supermajority in groups E1P and E2P, as shown in Table 6.

### 5.2. Similar Studies

There is a notable gap in the existing literature regarding the implicit analysis of individuals' awareness when sharing postings on social media potentially containing incidental data. To the best of our knowledge, there exists no study we can compare with, while not directly addressing this specific aspect, the research conducted by Padyab et al. [29] and Amon et al. [28] are interesting to approximate with. The study by Amon et al. [28] focuses on the psychological motivations behind users posting pictures of others, whereas the research by Padyab et al. [29] confronts participants with data extraction tools on their own social media profiles.

### 5.3. Statistical Significance

This study uses a confidence interval of 95%. The confidence interval reflects an estimated range of values. Furthermore, the confidence interval indicates the accuracy of the estimate. The margin of error is also used for statistical evaluation. In our study, the margin of error is 7.07%. This indicates the accuracy of the estimate in relation to an entire group. Altogether, the results of the study of 192 students reflect the opinions and awareness of Austrian students. Equation (1) represents the formula used for the margin of error *E* with *Standard Error of the Proportion* (SEP) and *Finite Population Correction* (FPC).

$$E = Z \times \sqrt{\frac{p \times (1 - p)}{n}} \times \sqrt{\frac{N - n}{N - 1}} = 7.07 \tag{1}$$

Here, *E* is the margin of error, *Z* is the Z-score associated with the desired confidence level, *p* as the estimated proportion of the population is set to 50%, *n* is the sample size, and *N* is the population size. The Z-score was set at 1.96. The target population are Austrian students, with a population of 288,381 as of February 2022 [38,39]. The population size is negligible because the FPC is 0.9999.

### 5.4. Trustability of Survey Results

All responses were received from students who received bonus points towards their grades. Due to data protection and ethics, the survey was designed to be 100% anonymous. Within the course, students had to enter one or multiple UUID tokens into the university submission system in order to receive offered bonus points. This token also had to be entered into the token collection surveys IDS2301U or IDS2302U. As a student, it was not allowed to have multiple tokens in the IDS2301U survey. However, IDS2302U received multiple entries since this was also the token submission survey for students who attended

three courses. Further, we are able to analyze the data of the token collection survey IDS2301U, IDS2302U, and the university submission system.

In order to understand why we are highly certain students did not take a survey twice, we need to describe the process in more detail. The survey and bonus point granting workflow is also visualized within Figure 5. Emails with the link to the survey IDS2301 were sent out alongside with emails to students who attended one or more classes. Students had to go through the survey to find the link for the token collection survey IDS2301U. Students had to generate the UUID token by themself and enter it into IDS2301U. Since IDS2301U was a two-question survey (email for updates and token) it would have been easier for students to ask their peers and simply enter their UUID, hence claiming that they had done the survey (IDS2301) rather than actually going through the survey (IDS2301). An analysis of the university data and the token collection survey showed that zero students claimed bonus points for multiple courses in IDS2301U. It is important to mention that not all enrolled students were graded and that not all graded students needed bonus points since the link to the survey was sent out close to the end of the course. Hence, students could estimate if an excellent grade was already reached or not, thus rendering bonus points useless.

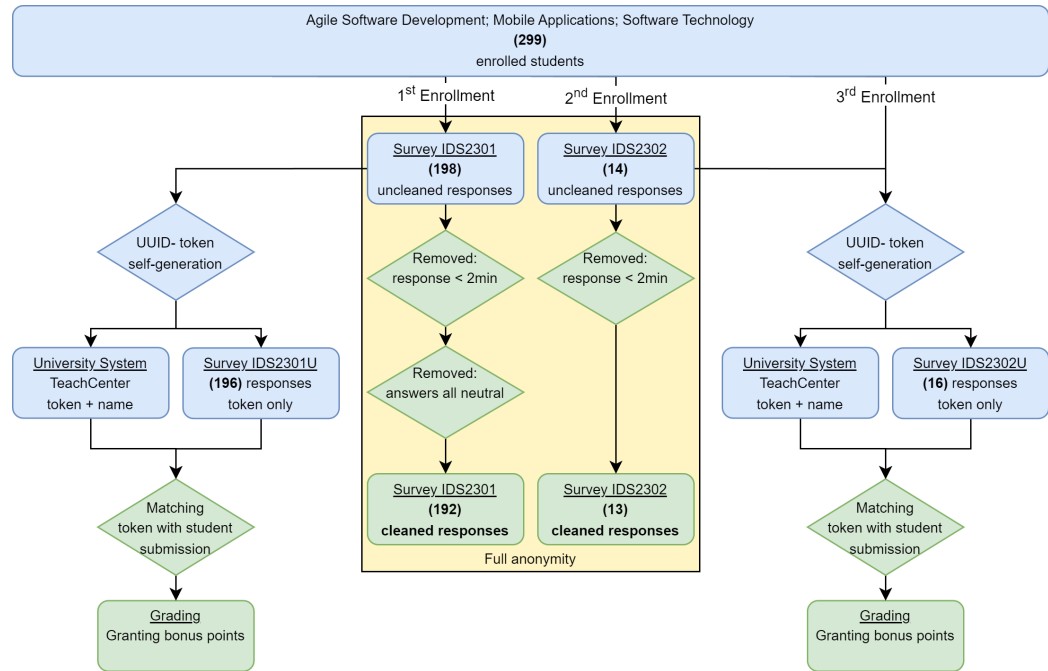

**Figure 5.** Depicts the workflow a student must undertake in order to receive bonus points, how anonymity is preserved, and how data are kept clean and trustworthy. Blue indicates a student action, whereas green indicates a lecturer or researcher role.

Students who enrolled to two or more courses received, alongside the IDS2301 survey link, the information and link to the secondary IDS2302 survey. The IDS2302 survey shows the same postings but asks what data and information they find in an open question. At the end a link to the dedicated token collection survey IDS2302U was revealed.

## 6. Conclusions

Disclosure of private information is crucial to social interactions, yet the awareness of privacy-compromising data hidden within in a self-disclosed posting needs more attention. This study extends the previous study of Kutschera [6] and using the comprehensively analyzed postings as seen in Figures 2 and 3. The implicit design of survey questions allowed us to gain inside information on the awareness of people about private data. Furthermore, our study shows that awareness about incidental data is very low, and this constitutes a

privacy and security concern. Our survey shows clear privacy concerns on data types full name (Q3.2), full address (Q3.4), information on relatives (Q3.8), phone number (Q3.9), parcel number of the property (Q3.12), security measures against burglars (Q3.19), as well as the absence thereof (Q3.20). Though participants were not forced to a decision by responding neutral on the 5-P Likert scale, 14.06% to 21.88% responded that, despite their privacy concerns, they were surprisingly willing to publish a posting, knowingly or not, that contains information considered privacy-compromising, thus incidental data.

Even though our survey achieved a confidence interval of 95%, the margin of error of 7.07% is still above the standard of 5% with 192 responses. Further, the results of our survey are limited with regard to interpretation, as the survey only asked Austrian students. With that in mind, we recommend a more widespread survey on the privacy and security issues of incidental data. Policymakers should also be made aware of these issues so that they can implement guidelines or other mechanisms that are latent to either raise awareness among the general public or alert persons before posting potentially harmful postings.

**Author Contributions:** Conceptualization, S.K., W.S. and H.D.; data curation, S.K. and P.R.; formal analysis, S.K.; investigation, S.K.; methodology, S.K. and H.D.; project administration, S.K.; resources, W.S.; software, S.K. and P.R.; supervision, W.S. and H.D.; validation, P.R., S.G. and P.D.; visualization, S.K. and P.R.; writing—original draft, S.K., P.R., S.G. and P.D.; writing—review and editing, S.K., W.S., P.R., S.G., P.D. and H.D. All authors have read and agreed to the published version of the manuscript.

**Funding:** The article processing charges (APC) was funded by Open Access Funding of Graz University of Technology.

**Institutional Review Board Statement:** Ethical review and approval were waived for this study due to Article 6 and Article 89 General Data Protection Regulation (EU) 2016/679 (GDPR) and further Section 7 Datenschutzgesetz (DSG) BGBl. I Nr. 165/1999 as amended BGBl. I Nr. 120/2017.

**Informed Consent Statement:** Informed consent was obtained from all subjects involved in the study.

**Data Availability Statement:** The data presented in this study are available upon reasonable request from the corresponding author.

**Conflicts of Interest:** The authors declare no conflicts of interest. The sponsors had no role in the design of the study; in the collection, analyses, or interpretation of data; in the writing of the manuscript, or in the decision to publish the results.

## Abbreviations

The following abbreviations are used in this manuscript:

| | |
|---|---|
| API | Application Programming Interface |
| ENF | Electric Network Frequency |
| FPC | Finite Population Correction |
| GDPR | General Data Protection Regulation (EU) 2016/679 |
| MDPI | Multidisciplinary Digital Publishing Institute |
| OSINT | Open Source Intelligence |
| SEP | Standard Error of the Proportion |
| UUID | Universal Unique Identifier |

## Appendix A. Survey Consent and Information

In order to inform participants and allow them to give their consent, an email (shown in the quote below) was sent out to all participants.

*Appendix A.1. Email to All 299 Enrolled Students*

> Dear students,
> We're excited to finally present you the opportunity to earn 10% bonus points for your current coursework.
> Our research will investigate privacy on social media through a survey. As per

the study design, the survey shall only be taken once**. The survey is anonymous and single responses cannot be traced back. The collected data will be analyzed, and results may be published in a journal or at a conference.

"IDS2301": *https://…*

Your feedback is valuable to our research, and we want to reward your time and effort. In order to gain the extra 10%, you need to drop a token into TeachCenter. This token is a randomly generated UUID and submitted to us in two ways. (This token concept won't affect the anonymity).

**For those attending two or more lectures: You will get an additional email with information on how to gain 10% for each of your courses. Hint: Follow-up surveys.
Thank you in advance for your participation and valuable feedback.
Kind Regards,
Stefan

*Appendix A.2. Email to Students Who Are Enrolled in Multiple Courses*

Dear students, you receive this email because you are enrolled in more than 1 course (ASD/ST or MA). Wwe're excited to finally present you the opportunity to earn 10% bonus points for another course with us. Our research will investigate privacy on social media through a survey. As per the study design, the survey shall only be taken once. The survey is anonymous and single responses cannot be traced back. The collected data will be analyzed, and results may be published in a journal or at a conference. It is important for the study design that the first survey with the ID "IDS2301" is filled out first (previous email), as they rely on each other.

"IDS2302": *https://…*

Your feedback is valuable to our research, and we want to reward your time and effort. In order to gain the extra 10%, you need to drop a token into TeachCenter. This token is a randomly generated UUID and submitted to us in two ways. (This token concept won't affect the anonymity).

Thank you in advance for your participation and valuable feedback.
Kind Regards,
Stefan

**Table A1.** Overview of surveys used in this study.

| ID | Motivation |
| --- | --- |
| IDS2301 | Main survey with 14 questions. First presentation of pictures to students. |
| IDS2301U | In case IDS2301 was answered: update survey for bonus point token and optionally the possibility to stay informed about this study. |
| IDS2302 | In case students needed to earn bonus points for more than one lecture, another survey that has open-ended questions regarding the pictures already seen from IDS202301. This survey has the sole purpose of preventing students from multi-answering IDS202301 and, therefore, not influencing the results as a whole. |
| IDS2302U | In case IDS2302 was answered: update survey for bonus point token and optionally the possibility to stay informed about this study. |

## Appendix B. IDS2301 Survey Questions & Results

**Table A2.** Shows the different social media platforms usages and the respective behavior of uploading content (UC) and consuming content (CC). Cell background is based on the value of the cell.

| N | | UC | CC | Facebook | YouTube | Reddit | Twitter | 4Chan | Mastodon | Tumblr | Instagram | TikTok | Snapchat | Discord | Pinterest | LinkedIn | Xing |
|---|---|---|---|---|---|---|---|---|---|---|---|---|---|---|---|---|---|
| 192 | ALL | 2.10 | 4.71 | 2.06 | 4.84 | 2.75 | 2.10 | 1.07 | 1.17 | 1.11 | 4.25 | 2.86 | 2.80 | 4.30 | 1.62 | 2.20 | 1.28 |
| 16 | ALP | 2.56 | 5.19 | 2.15 | 5.12 | 3.00 | 1.91 | 1.29 | 1.00 | 1.00 | 4.19 | 2.91 | 3.00 | 4.94 | 1.50 | 2.23 | 1.17 |
| 88 | ANP | 1.84 | 4.47 | 1.89 | 4.65 | 2.75 | 1.89 | 1.00 | 1.04 | 1.08 | 3.82 | 2.43 | 2.59 | 4.28 | 1.49 | 2.00 | 1.21 |
| 27 | E1P | 2.19 | 5.04 | 2.19 | 4.93 | 3.10 | 1.94 | 1.20 | 1.00 | 1.00 | 4.38 | 2.41 | 2.94 | 4.81 | 1.38 | 2.65 | 1.40 |
| 42 | E2P | 2.57 | 5.14 | 2.19 | 4.93 | 2.84 | 2.17 | 1.24 | 1.27 | 1.20 | 4.85 | 3.14 | 3.18 | 4.50 | 1.77 | 2.30 | 1.33 |
| 73 | BY | 2.15 | 4.64 | 2.07 | 4.58 | 2.82 | 1.83 | 1.00 | 1.10 | 1.16 | 4.46 | 2.74 | 2.56 | 4.26 | 1.33 | 2.26 | 1.19 |
| 119 | BN | 2.07 | 4.76 | 2.05 | 4.99 | 2.71 | 2.25 | 1.11 | 1.21 | 1.09 | 4.14 | 2.92 | 2.92 | 4.32 | 1.81 | 2.16 | 1.32 |
| 136 | LR | 2.16 | 4.74 | 2.14 | 4.78 | 2.71 | 2.03 | 1.11 | 1.22 | 1.14 | 4.34 | 3.06 | 2.83 | 4.27 | 1.71 | 2.25 | 1.33 |
| 50 | LO | 1.92 | 4.60 | 1.62 | 4.98 | 2.79 | 2.38 | 1.00 | 1.06 | 1.06 | 3.97 | 2.46 | 2.84 | 4.49 | 1.19 | 1.96 | 1.17 |
| 119 | AS | 2.21 | 4.87 | 2.04 | 4.78 | 2.73 | 2.21 | 1.12 | 1.21 | 1.15 | 4.44 | 2.97 | 3.11 | 4.42 | 1.77 | 1.89 | 1.20 |
| 67 | AB | 1.91 | 4.40 | 2.02 | 4.89 | 2.85 | 1.97 | 1.00 | 1.11 | 1.06 | 3.86 | 2.77 | 2.26 | 4.11 | 1.23 | 2.67 | 1.43 |
| 29 | BYS | 2.52 | 5.00 | 2.09 | 4.63 | 2.94 | 1.69 | 1.00 | 1.00 | 1.25 | 4.74 | 2.93 | 3.35 | 4.30 | 1.67 | 1.81 | 1.11 |
| 90 | BNS | 2.11 | 4.82 | 2.02 | 4.83 | 2.67 | 2.37 | 1.16 | 1.28 | 1.12 | 4.34 | 2.98 | 3.04 | 4.46 | 1.81 | 1.92 | 1.23 |
| 86 | LRS | 2.27 | 4.85 | 2.13 | 4.72 | 2.75 | 1.94 | 1.15 | 1.24 | 1.21 | 4.44 | 2.93 | 3.02 | 4.40 | 1.92 | 1.83 | 1.20 |
| 32 | LOS | 2.06 | 4.88 | 1.64 | 4.94 | 2.55 | 3.06 | 1.00 | 1.12 | 1.00 | 4.56 | 3.06 | 3.40 | 4.41 | 1.27 | 2.06 | 1.20 |
| 41 | BYB | 1.93 | 4.34 | 2.00 | 4.49 | 2.88 | 2.06 | 1.00 | 1.18 | 1.10 | 4.27 | 2.64 | 2.00 | 4.17 | 1.12 | 2.54 | 1.27 |
| 26 | BNB | 1.88 | 4.50 | 2.05 | 5.50 | 2.83 | 1.88 | 1.00 | 1.00 | 1.00 | 3.29 | 2.92 | 2.60 | 4.00 | 1.40 | 2.93 | 1.60 |
| 49 | LRB | 1.98 | 4.53 | 2.08 | 4.87 | 2.68 | 2.24 | 1.00 | 1.18 | 1.00 | 4.18 | 3.53 | 2.38 | 4.06 | 1.29 | 3.00 | 1.62 |
| 16 | LOB | 1.69 | 3.94 | 1.60 | 5.00 | 3.25 | 1.33 | 1.00 | 1.00 | 1.12 | 2.40 | 1.33 | 2.00 | 4.50 | 1.11 | 1.82 | 1.12 |
| 52 | S250 | 2.02 | 4.62 | 1.68 | 4.84 | 2.71 | 2.18 | 1.00 | 1.06 | 1.00 | 3.98 | 2.85 | 2.89 | 4.50 | 1.48 | 1.90 | 1.22 |
| 134 | B250 | 2.12 | 4.72 | 2.14 | 4.82 | 2.76 | 2.10 | 1.11 | 1.22 | 1.18 | 4.37 | 2.94 | 2.82 | 4.25 | 1.62 | 2.28 | 1.32 |
| 161 | SCC | 2.23 | 5.11 | 2.07 | 4.96 | 2.80 | 2.22 | 1.04 | 1.07 | 1.07 | 4.44 | 3.04 | 2.94 | 4.27 | 1.69 | 2.20 | 1.26 |
| 21 | SUC | 4.52 | 5.38 | 2.58 | 4.45 | 3.00 | 2.43 | 1.33 | 1.80 | 2.00 | 5.38 | 4.47 | 3.94 | 3.90 | 2.64 | 2.87 | 1.83 |
| 156 | M | 1.94 | 4.56 | 1.96 | 4.96 | 2.74 | 2.16 | 1.08 | 1.19 | 1.09 | 4.08 | 2.48 | 2.71 | 4.40 | 1.34 | 2.19 | 1.28 |
| 32 | F | 2.94 | 5.44 | 2.50 | 4.30 | 2.67 | 1.82 | 1.00 | 1.00 | 1.25 | 5.14 | 4.39 | 3.22 | 3.80 | 2.56 | 2.22 | 1.25 |

**Table A3.** Survey IDS2301 Questions 6; social media usage.

| Reference | Question | Response Type |
|---|---|---|
| Q6 | Reflecting on your social media behavior. How often do you… | |
| Q6.1 | … upload content on social media? | 6-P Likert Scale |
| Q6.2 | … consume content on social media? | 6-P Likert Scale |

**Table A4.** Survey IDS2301 Questions 7; social media platforms.

| Reference | Question | Response Type |
|---|---|---|
| Q7 | What social media platforms do you use? 0 [No answer], 1 [Never] to 6 [Very Frequently] | |
| Q7.1 | Facebook | 6-P Likert Scale |
| Q7.2 | YouTube | 6-P Likert Scale |
| Q7.3 | Reddit | 6-P Likert Scale |
| Q7.4 | Twitter | 6-P Likert Scale |
| Q7.5 | 4Chan | 6-P Likert Scale |
| Q7.6 | Mastodon | 6-P Likert Scale |
| Q7.7 | Tumblr | 6-P Likert Scale |
| Q7.8 | Instagram | 6-P Likert Scale |
| Q7.9 | TikTok | 6-P Likert Scale |
| Q7.10 | Snapchat | 6-P Likert Scale |
| Q7.11 | Discord | 6-P Likert Scale |
| Q7.12 | Pinterest | 6-P Likert Scale |
| Q7.13 | LinkedIn | 6-P Likert Scale |
| Q7.14 | Xing | 6-P Likert Scale |

**Table A5.** Survey IDS2301 Questions 8–14; Demographic.

| Reference | Question | Response Type |
|---|---|---|
| Q8 | Your gender | Single choice |
| Q9 | Your age | Number field |
| Q10 | What describes your current main study the most? | Single choice |
| Q11 | Are you holding a bachelor's degree? | Boolean |
| Q12 | Do you currently live in | |
| Q12.1 | a rented house or flat? | Boolean |
| Q12.2 | an owned house or flat? | Boolean |
| Q13 | How many people live in your town/city? | Single choice |
| Q14 | Anything else you want to tell me? | open-ended |

**Table A6.** Shows the data types from Question 3 (Q3.1–Q3.20). In rows *Example 1* (E.1) and *Example 2 (E.2)* the data types found are marked with *Yes (X)* and *No (O)*. Column *N* states the number of participants in each row. Column *Abbr.* lists all subgroups using their filter identifier as explained in Table 4. The cells represent the percentage of participants of the said subgroup who responded with either ***disagree* or *strongly disagree*** on questions about the privacy compromise regarding data types Q3.1–Q3.20.

| N | Abbr. | 1 | 2 | 3 | 4 | 5 | 6 | 7 | 8 | 9 | 10 | 11 | 12 | 13 | 14 | 15 | 16 | 17 | 18 | 19 | 20 |
|---|---|---|---|---|---|---|---|---|---|---|---|---|---|---|---|---|---|---|---|---|---|
| | E.1 | X | X | O | X | X | O | O | X | O | O | O | X | X | X | O | X | X | O | X | X |
| | E.2 | X | X | O | X | O | O | O | X | X | O | O | X | X | X | O | X | O | O | X | X |
| 192 | ALL | 52.08 | 18.75 | 41.67 | 6.25 | 44.27 | 48.96 | 7.81 | 14.58 | 8.85 | 34.38 | 27.60 | 21.88 | 53.65 | 53.12 | 19.27 | 57.29 | 51.56 | 35.42 | 19.27 | 17.19 |
| 16 | ALP | 56.25 | 0.00 | 18.75 | 12.50 | 56.25 | 56.25 | 6.25 | 0.00 | 6.25 | 31.25 | 31.25 | 18.75 | 37.50 | 62.50 | 18.75 | 50.00 | 43.75 | 56.25 | 37.50 | 31.25 |
| 88 | ANP | 51.14 | 17.05 | 37.50 | 4.55 | 39.77 | 52.27 | 7.95 | 18.18 | 9.09 | 32.95 | 25.00 | 20.45 | 46.59 | 37.50 | 15.91 | 50.00 | 53.41 | 27.27 | 15.91 | 18.18 |
| 27 | E1P | 48.15 | 11.11 | 33.33 | 11.11 | 48.15 | 48.15 | 3.70 | 3.70 | 11.11 | 29.63 | 25.93 | 18.52 | 44.44 | 66.67 | 14.81 | 48.15 | 37.04 | 51.85 | 29.63 | 18.52 |
| 42 | E2P | 54.76 | 14.29 | 35.71 | 9.52 | 52.38 | 47.62 | 7.14 | 9.52 | 7.14 | 30.95 | 28.57 | 21.43 | 57.14 | 61.90 | 16.67 | 66.67 | 50.00 | 50.00 | 23.81 | 16.67 |
| 73 | BY | 57.53 | 20.55 | 39.73 | 4.11 | 47.95 | 46.58 | 8.22 | 10.96 | 6.85 | 38.36 | 34.25 | 21.92 | 52.05 | 52.05 | 19.18 | 49.32 | 45.21 | 41.10 | 20.55 | 20.55 |
| 119 | BN | 48.74 | 17.65 | 42.86 | 7.56 | 42.02 | 50.42 | 7.56 | 16.81 | 10.08 | 31.93 | 23.53 | 21.85 | 54.62 | 53.78 | 19.33 | 62.18 | 55.46 | 31.93 | 18.49 | 15.13 |
| 136 | LR | 48.53 | 19.85 | 41.18 | 6.62 | 51.47 | 44.12 | 7.35 | 16.91 | 10.29 | 36.76 | 28.68 | 23.53 | 55.15 | 52.21 | 23.53 | 56.62 | 53.68 | 41.18 | 16.91 | 19.12 |
| 50 | LO | 58.00 | 14.00 | 46.00 | 6.00 | 30.00 | 60.00 | 10.00 | 10.00 | 6.00 | 32.00 | 28.00 | 18.00 | 48.00 | 54.00 | 10.00 | 56.00 | 46.00 | 24.00 | 26.00 | 14.00 |
| 119 | AS | 52.94 | 19.33 | 45.38 | 7.56 | 42.86 | 50.42 | 8.40 | 13.45 | 7.56 | 38.66 | 23.53 | 22.69 | 53.78 | 49.58 | 23.53 | 59.66 | 52.94 | 34.45 | 18.49 | 19.33 |
| 67 | AB | 50.75 | 17.91 | 34.33 | 4.48 | 47.76 | 44.78 | 7.46 | 16.42 | 10.45 | 26.87 | 32.84 | 20.90 | 53.73 | 59.70 | 13.43 | 55.22 | 49.25 | 37.31 | 16.42 | 11.94 |
| 29 | BYS | 58.62 | 24.14 | 48.28 | 6.90 | 48.28 | 44.83 | 10.34 | 6.90 | 6.90 | 48.28 | 27.59 | 20.69 | 48.28 | 34.48 | 24.14 | 55.17 | 51.72 | 41.38 | 20.69 | 24.14 |
| 90 | BNS | 51.11 | 17.78 | 44.44 | 7.78 | 41.11 | 52.22 | 7.78 | 15.56 | 7.78 | 35.56 | 22.22 | 23.33 | 55.56 | 54.44 | 23.33 | 61.11 | 53.33 | 32.22 | 17.78 | 17.78 |
| 86 | LRS | 50.00 | 22.09 | 44.19 | 6.98 | 50.00 | 46.51 | 6.98 | 16.28 | 8.14 | 41.86 | 26.74 | 24.42 | 56.98 | 48.84 | 27.91 | 58.14 | 53.49 | 38.37 | 18.60 | 23.26 |
| 32 | LOS | 59.38 | 12.50 | 50.00 | 9.38 | 25.00 | 59.38 | 12.50 | 6.25 | 6.25 | 31.25 | 15.62 | 18.75 | 46.88 | 50.00 | 12.50 | 62.50 | 53.12 | 25.00 | 18.75 | 9.38 |
| 41 | BYB | 58.54 | 19.51 | 29.27 | 2.44 | 46.34 | 46.34 | 7.32 | 12.20 | 4.88 | 29.27 | 34.15 | 21.95 | 56.10 | 63.41 | 17.07 | 48.78 | 41.46 | 39.02 | 14.63 | 14.63 |
| 26 | BNB | 38.46 | 15.38 | 42.31 | 7.69 | 50.00 | 42.31 | 7.69 | 23.08 | 19.23 | 23.08 | 30.77 | 19.23 | 50.00 | 53.85 | 7.69 | 65.38 | 61.54 | 34.62 | 19.23 | 7.69 |
| 49 | LRB | 46.94 | 16.33 | 34.69 | 6.12 | 53.06 | 38.78 | 8.16 | 16.33 | 12.24 | 26.53 | 30.61 | 20.41 | 51.02 | 57.14 | 16.33 | 55.10 | 53.06 | 44.90 | 12.24 | 10.20 |
| 16 | LOB | 56.25 | 18.75 | 31.25 | 0.00 | 37.50 | 62.50 | 6.25 | 18.75 | 6.25 | 31.25 | 43.75 | 18.75 | 56.25 | 62.50 | 6.25 | 50.00 | 37.50 | 18.75 | 31.25 | 18.75 |
| 52 | S250 | 46.15 | 19.23 | 48.08 | 3.85 | 48.08 | 61.54 | 9.62 | 9.62 | 3.85 | 30.77 | 23.08 | 23.08 | 48.08 | 51.92 | 23.08 | 55.77 | 63.46 | 40.38 | 25.00 | 17.31 |
| 134 | B250 | 53.73 | 17.91 | 39.55 | 7.46 | 44.03 | 43.28 | 7.46 | 16.42 | 10.45 | 36.57 | 29.85 | 21.64 | 55.22 | 52.99 | 18.66 | 57.46 | 46.27 | 34.33 | 16.42 | 17.16 |
| 161 | SCC | 55.28 | 19.25 | 42.86 | 5.59 | 44.72 | 47.83 | 6.83 | 13.04 | 8.07 | 35.40 | 25.47 | 21.12 | 55.90 | 55.28 | 21.74 | 57.14 | 52.80 | 37.89 | 20.50 | 17.39 |
| 21 | SUC | 42.86 | 42.86 | 47.62 | 14.29 | 66.67 | 42.86 | 19.05 | 4.76 | 9.52 | 33.33 | 33.33 | 38.10 | 57.14 | 52.38 | 23.81 | 47.62 | 42.86 | 28.57 | 23.81 | 28.57 |
| 156 | M | 47.44 | 17.95 | 43.59 | 6.41 | 43.59 | 48.72 | 8.97 | 17.31 | 9.62 | 35.26 | 30.13 | 23.08 | 53.21 | 55.77 | 22.44 | 56.41 | 51.92 | 36.54 | 20.51 | 17.31 |
| 32 | F | 75.00 | 25.00 | 37.50 | 6.25 | 50.00 | 43.75 | 3.12 | 3.12 | 6.25 | 31.25 | 18.75 | 15.62 | 56.25 | 37.50 | 6.25 | 62.50 | 53.12 | 31.25 | 12.50 | 15.62 |

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
