# Peer review of "Incidental Data: A Survey towards Awareness on Privacy-Compromising Data Incidentally Shared on Social Media"

_jcp, doi:10.3390/jcp4010006_

Round 1

Reviewer 1 Report

Comments and Suggestions for Authors

The paper intends to study privacy awareness of peoples in social media with incidental data by Q3.n-Q7.x, Q8-Q14. So much questions are too complex to understand. You should design questions surrounding your motivation. Ay the end, you should present some significant insights through such experiments. And the expression is so intricate and some paragraghs are repeated.

Author Response

Thank you for the valuable feedback. We addressed the comment by adding a new paragraph in summarizing significant insights and a visual representation within the new Table 9 “[...]Shows excerpt of data types from question three[..]”. As for the repeated paragraphs, this was a sincere slip-up error during transitioning the professional proofreader's response we got as MS-word files into LaTeX. Lines 27-41 were corrected by proofreaders, whereas 42-55 are outdated. We revised the abstract to also focus more on significant results and removed unnecessary details.  

Reviewer 2 Report

Comments and Suggestions for Authors

This paper performs a survey about awareness on privacy compromising data on social media based on 194 students' opinions by asking the questions regarding postings contained incidental data.

In the background, the authors explained very well the reason why this survey is important based on several perspectives of related previous research reports.

However, the new idea, motivation, advantages, merit, and contribution of the proposed tehcnique are still not clearly visible. Although, the authors extended the previous research method, Kutschera [5] by combining findings regarding data types of present study shown in Table 2.

The recommendation to the authors is to make clearer the new idea, advantages, merit, and contribution.

Moreover, comparing the proposed survey technique to other similar existing surveys is very important to prove the contribution of the proposed technique. Apart from that, it should also be compared with the results of implementation, experiment, measurement and evaluation with other similar existing surveys.

In addition, Lines 36-41 are repeated into Lines 50-55.

Author Response

Thank you for the valuable feedback. 

We addressed the comment, to make clearer the new idea, advantages, merit, and contribution, by massively restructuring the Section Introduction as well as revising the Abstract. Further, Section 3, Implementation, now focuses on the main survey IDS2301, whereas the other surveys are repositioned with an explanation and a new workflow image in the Section “Trustability of Data”. 

To the best of our knowledge, there exists a significant gap in the literature that focuses on the awareness of self-disclosed privacy-compromising social media postings. However, we added a new Section 2.3 “Privacy from an Awareness Perspective” with related studies that, although not exactly the same, share a common theme with our research. Further, they are discussed in the Section 5.2.

As for the repeated paragraphs, the paper has received professional proofreading through an external service offered by the university. During the hideous implementation from MS Word to LaTeX, a slip-up mistake happened where paragraphs appeared multiple times within the submitted paper, L27-55 of the initial submission (L27-41 proofreaders-version; L42-55 original-version).  We sincerely apologize for that embarrassing mistake.

Reviewer 3 Report

Comments and Suggestions for Authors

Dear authors,

I checked your manuscript in detail and have several comments. Please, take them into account to improve the work done.

Comment 1. Please, take additional time to prepare the final version of the manuscript. 

The paper is not well-written and well-organized, contains typos and requires to continuously check multiple tables to not miss the necessary context. Thus, some reorganization is necessary to improve the readability.

Moreover, last 3 paragraphs in the Introduction section are written 2 times.

Comment 2. Please, reveal the novelty of the work done and your contribution to the research field.

It is not clear what is the novelty of the work done. It looks like already know questions were partially taken from GDPR and given to students. After that, the survey results were checked using well-known techniques.

Are there other researches like that? Is it possible to compare results? What are the main insights of your study? Are they complementing other research in the area or not?

Comment 3. Is it possible to trust the results of the research?

It is mentioned, that there was a theoretical possibility for the student to complete the survey multiple times, and the authors only believe that nobody did it.

Moreover, it is not mentioned if the results of time spent on a survey were used to drop suspicious results (for example, with very short difference between start and end time).

Author Response

Thank you for the valuable feedback and your recommendation to take additional time, as we believe this made it easier to ask for an extension of the initially scheduled 10-day deadline.

Ad Comment 1: 

Dependant tables, such as Tables 4 & 5, as well as Tables 6 & 7, are now re-structured. More specifically, Tables 4 and 6 (listing of questions) from the initial manuscript are now merged into a single table, namely Table 4 in the revised manuscript. This change further underlines that those privacy measurements are asked twice but within a different context. Table 5 and 7 from the initial manuscript now reside on a single page. A new table puts important numbers as an excerpt from Table 2, including significant results from Table 1 and Table 3, into one --> Table 13. Section "Implementation" was restructured and divided into subsections "Recruitment", "Assessment Design", and "Questions". Another restructuring was done, as we renamed Section "Evaluation" into "Discussion" and former Section "Discussion" into "Conclusion" with restructuring and rephrasing the context in order to fit the section header. 

Moreover, we removed noise from the manuscript as not strictly necessary tables and information were removed into the Appendix but referred to within the text. 

The new Figure 1, showing a Venn-diagram, shall improve readability by visualising multiple paragraphs from the section "Implementation" and further adding new information on how many students were enrolled into multiple courses –> overlapping section within Venn-diagram (19 students)

The paper has received professional proofreading through an external service offered by the university. During the hideous implementation from MS-Word to LaTeX, a slip-up mistake happened where paragraphs appeared multiple times within the submitted paper, L27-55 of the initial submission (L27-41 proofreaders-version; L42-55 original-version).  We sincerely apologize for that embarrassing mistake.

Ad Comment 2: 

Thank you for pointing that out. In fact, questions are not taken from GDPR but certain information may enjoy greater protection. To be specific: blood type (Q3.6), as generally asked within the survey, enjoys greater protection under Art. 9 GDPR. We do not dare to answer how the blood type of a person may be used to invade one's privacy; however, since categorized under GDPR we too addressed it as sensitive personal information. However, neither of the presented social media pictures contained or led to that sort of information. 

We revised the phrasing to be more specific and clearer. (L206-L208, revised manuscript)

To the best of our knowledge, there exists a significant gap in the literature that focuses on the awareness of self-disclosed privacy-compromising social media postings. However, we added a new Section 2.3 “Privacy from an Awareness Perspective” with related studies that, although not exactly the same, share a common theme with our research. Further, they are discussed in the Section 5.2.

Ad Comment 3: 

This is a very important question. To our luck, we have robust data available since we know which student requested bonus points for which survey. That allows us to point out students who either took the IDS2301 survey multiple times or simply requested multiple bonus points for the same survey. Luckily, no such case is present. 

We revised the document and explained this in a newly added Section “Trustability of Survey Results”. Figure 5 adds a chart to help readers visualize the described process. 

Based on the raised concern, we reassessed responses based on their duration. We found that, in general participants took less time than expected and saw that responses under two minutes were garbage but responses at or above two minutes had already complete and meaningful results including short optional open-text explainations. We have removed all five responses (of which three were already excluded) beneath that threshold and updated every table and text reference accordingly. Removing those responses did not change any findings, as, for instance, all one-half majorities or two-thirds super majorities stood the same. Instead of 194 we now have 192 responses. 

Round 2

Reviewer 1 Report

Awareness on Privacy Compromising Data Incidentally Shared on Social Media is very important. The Authors conduct a questionnaire survey of 192 students. The topic is valuable. But the expression should be polished.

1, you mention 4 survey, so you should show the relationship between them, also relationship between these surveys and questions. Certainly, you may depict them in a picture.

2, you should show your questions altogether.

3, In Table 5,6,7, you should indicate the meanings of the values and describe how to get them. 

Author Response

Ad introduction:

We revised the introduction and explained the concerned parts in more detail.  

Ad methods:

Table 4 contains the merged information from previous round two tables. This was changed in round 1 in order to reduce noise and enhance readability. We added an explanation in the text as well as in the table description.

Ad results:

We clarified the meaning of the values from Table 5,6,7 within the Section Results. They are now not only described within the table description but also within the main text. 

Ad 1-3

  1. We resolved this comment by referring to the related section where we explain the relation in more detail where we introduce and mention those 4 surveys for the first time. 
  2. Currently questions 1 to 14 are represented within Tables 1, 2 and 4 in the main mansucript and A3, A4, and A5 within the Appendix. Generating an altogether-overview of all questions within the main manuscript would add a lot of noise to it. On the contrary, putting all into the Appendix disrupts the clarity and structure of the manuscript as changed by recommendations we received in the first round. An additional altogether-overview in the Appendix may also disturb the reader flow due to duplications.
  3. We clarified the meaning of the values and now describe them not only in the table description but also within the main text.  

Reviewer 2 Report

The authors have already addressed to make clearer the idea by restructuring the Introduction and revising the Abstract. The Section of Implementation now focuses on the main survey IDS2301, whereas the other surveys are repositioned with an explanation. 

The authors also inserted a new Section 2.3 “Privacy from an Awareness Perspective” with related studies and discussion in the Section 5.2.

However, a comparison between the results of this work and similar existing works would make this manuscript much better.

The authors have already addressed to make clearer the idea by restructuring the Introduction and revising the Abstract. The Section of Implementation now focuses on the main survey IDS2301, whereas the other surveys are repositioned with an explanation. 

The authors also inserted a new Section 2.3 “Privacy from an Awareness Perspective” with related studies and discussion in the Section 5.2.

However, a comparison between the results of this work and similar existing works would make this manuscript much better.

Author Response

Thank you for your feedback.

Correct, Section 2.3 introduces studies we found most applicable and Section 5.2, stating that we cannot present any suitable studies, puts them into some sort of context with our study. On the one hand, the results of Amon (2023) reflect privacy awareness on posted content about others and evaluate their psychological dark triad profile. On the other hand, Padyab (2019) used an automated data extraction tool within focus groups. The results are interesting and show that users are unaware of some information the data extraction tool discovered. However, the methodology is different as Padyab (2019) used qualitative methods, were we used quantitative methods. Further, there is no prevention on influence on participants but a short teaching on data mining. We reduced the influence on direct questions  (L24-26 & L90-92) . 

We updated the Section 5.2 for more clarity.

Reviewer 3 Report

With all improvements done, I believe, that the manuscript can be accepted in its current form.

In my first review, I made several comments about the content of the manuscript. My main advice was in the reconsideration of the manuscript's structure and improvement of its content.

More precisely, my comments were as follows:

1. Please, take additional time to prepare the final version of the manuscript.

2. Please, reveal the novelty of the work done and your contribution to the research field.

3. Is it possible to trust the results of the research?

Now, with the new version of the manuscript available, I can clearly state that all my comments were addressed. The authors did a great job in the improvement of the work done.

Author Response

Thank you again for your valuable recommendations. We considered them during this second round as well.